# Unlocking the Potential: Quercetin and Its Natural Derivatives as Promising Therapeutics for Sepsis

**DOI:** 10.3390/biomedicines12020444

**Published:** 2024-02-16

**Authors:** Tian Wang, Linxi Lv, Hui Feng, Wei Gu

**Affiliations:** 1Center of Smart Laboratory and Molecular Medicine, School of Medicine, Chongqing University, Chongqing 400030, China; wangtian211002@163.com (T.W.); l18523921569@163.com (L.L.); 2College of Bioengineering, Chongqing University, Chongqing 400044, China

**Keywords:** sepsis, quercetin and its derivatives, anti-inflammatory, antioxidant

## Abstract

Sepsis is a syndrome of organ dysfunction caused by an uncontrolled inflammatory response, which can seriously endanger life. Currently, there is still a shortage of specific therapeutic drugs. Quercetin and its natural derivatives have received a lot of attention recently for their potential in treating sepsis. Here, we provide a comprehensive summary of the recent research progress on quercetin and its derivatives, with a focus on their specific mechanisms of antioxidation and anti-inflammation. To obtain the necessary information, we conducted a search in the PubMed, Web of Science, EBSCO, and Cochrane library databases using the keywords sepsis, anti-inflammatory, antioxidant, anti-infection, quercetin, and its natural derivatives to identify relevant research from 6315 articles published in the last five years. At present, quercetin and its 11 derivatives have been intensively studied. They primarily exert their antioxidation and anti-inflammation effects through the PI3K/AKT/NF-κB, Nrf2/ARE, and MAPK pathways. The feasibility of these compounds in experimental models and clinical application were also discussed. In conclusion, quercetin and its natural derivatives have good application potential in the treatment of sepsis.

## 1. Introduction

Sepsis remains a life-threatening illness with high morbidity and mortality, especially in low- and middle-income regions [1,2]. It affects approximately 48.9 million individuals worldwide, leading to an estimated 11 million deaths annually [3]. These deaths are predominantly associated with organ dysfunction caused by an uncontrolled immune response to infections [4,5]. Although advances in therapeutic interventions and critical care have been substantial, the treatment of sepsis continues to be a major challenge, relying mainly on timely identification, the immediate administration of antibiotics, and supportive care. This highlights the urgent need for more effective treatment approaches to improve patient outcomes.

There has been increasing interest in exploring natural formulations and active ingredients as potential treatments for sepsis and its complications. For example, traditional Chinese medicines like Xuebijing and Huanglian–Jiedu decoctions were investigated for their potential to control sepsis during the COVID-19 pandemic [6,7]. Quercetin, a natural compound found in fruits and vegetables, has similarly attracted interest. Its anti-inflammatory effects have been widely supported by computational prediction, experimental verification, and clinical analysis [8]. Additionally, quercetin displays antioxidant properties and thus mitigates the harmful effects of sepsis [9]. Equally remarkable, quercetin has been gradually recognized for its antimicrobial effects against various bacteria as well as fungi and viruses, even drug-resistant ones [10]. These characteristics suggest that its application as a potent anti-sepsis agent could be highly favorable to the patient. During the past few years, its molecular mechanisms and pharmaceutical properties have also been extensively investigated [10,11]. Despite the existing literature on the anti-septic effects of quercetin and its derivatives, recent progress reviews and product information in this field are limited. By exploring the mechanisms and pathways involved in its treatment of sepsis, we propose that improving its bioavailability and solubilizing properties while retaining its natural advantages are valuable avenues for future research.

## 2. Inflammatory Pathways in Sepsis

It is now accepted widely that the occurrence of sepsis is multifactorial and involves dysregulated host responses to infection. For instance, lipopolysaccharides (LPS, important outer membrane components of Gram-negative bacteria) play a pivotal role in the development of sepsis [12]. In general, the interaction between LPS and Toll-like receptor 4 (TLR4) activates transcription factors like nuclear factor-kappa B (NF-κB), leading to the production of proinflammatory cytokines, including tumor necrosis factor-α (TNF-α), interleukin-1β (IL-1β), and IL-6. However, the excessive production of these cytokines can result in multiple organ dysfunction and fatal sepsis [13].

In addition to the disordered inflammatory response, an aberrant oxidative stress response also contributes to the pathogenesis of sepsis. The excessive release of reactive oxygen species (ROS) by immune cells in sepsis overwhelms the body’s antioxidant defense system, leading to oxidative stress and cellular damage. This oxidative stress state further activates the NF-κB signaling pathway, promoting the overproduction of inflammatory factors like TNF-α, IL-1β, and cyclooxygenase-2 (COX-2) [14,15]. The mitogen-activated protein kinase (MAPK) pathway is activated downstream of TLR4. MAPKs, including extracellular signal-regulated kinase (ERK), c-Jun N-terminal kinase (JNK), and p38 kinase, play crucial roles in the production of proinflammatory cytokines and the regulation of immune cell activation [16,17]. Moreover, the interferon regulatory factor 3 (IRF3) pathway contributes to the production of type I interferons and other proinflammatory cytokines [18].

Not surprisingly, excessive production of pro-inflammatory cytokines and an imbalanced inflammatory and oxidative stress response are known to contribute to the severity of sepsis. Addressing this issue, researchers have focused on the role of nuclear factor erythroid 2-related factor 2 (Nrf2), a crucial transcription factor involved in regulating the intracellular antioxidative stress response [19]. In situations of increased oxidative stress, Nrf2 dissociates from Kelch-like ECH-associated protein 1 (Keap1) and translocates into the nucleus. Once inside the nucleus, Nrf2 binds to antioxidant response elements (AREs) in the DNA, subsequently activating the expression of antioxidant and detoxifying enzymes. This activation process protects cells from sepsis-induced injury by mitigating oxidative stress, reducing inflammatory responses, and promoting cellular defense mechanisms against oxidative damage [20]. Furthermore, Nrf2 activation can inhibit the production of inflammatory factors such as IL-1β and TNF-α, thus reducing the inflammatory response and tissue damage [21]. Additionally, Nrf2 can regulate cell death processes such as apoptosis and autophagy, which impact the pathological process of sepsis [14,19]. Therefore, potential therapeutic interventions targeting the TLR4 signaling pathway and oxidative stress may hold promise in the treatment of sepsis.

## 3. Overview of Quercetin and Its Natural Derivatives

Quercetin and its natural derivatives, such as isoquercetin, rutin, and quercitrin, are known for their noteworthy biological activities attributed to their distinctive chemical structure (Figure 1). Quercetin exhibits antioxidant and anti-inflammatory properties owing to its flavonoid skeleton and numerous hydroxyl groups [10]. These qualities enable it to shield cells against oxidative damage and mitigate inflammation. They are ubiquitously present in plant-based foods, including fruits, vegetables, grains, and herbs [22,23]. Some of the richest sources of quercetin include apples, onions, berries, citrus fruits, grapes, tomatoes, broccoli, and leafy greens [23]. Efforts have been made to elucidate the bioactive potentials of quercetin and its derivatives. In fact, it is now widely appreciated that they exert a variety of beneficial effects, including antioxidant, anti-inflammatory, antitumor, antibacterial, immunosuppressive, and neuroprotective properties [24,25,26]. These properties not only make quercetin and its derivatives promising candidates for the treatment of various diseases, but they have also provided knowledge and understanding of how they and the presence of quercetin in natural foods benefit personal health.

Quercetin and its derivatives are polyphenolic substances that are widely present in nature. Solvent extraction is commonly used to extract such compounds by soaking plant materials in solvents (such as ethanol, methanol, etc.) [27]. High-performance liquid chromatography (HPLC) and other chromatographic techniques could be used for further separation and purification [28,29].

Nevertheless, quercetin is known for a low bioavailability associated with its poor solubility, stability, and absorption characteristics [30]. Accordingly, researchers have attempted to modify and generate novel quercetin derivatives with suitable pharmacological properties. Currently, the most extensively studied quercetin derivatives are glycosides and methyl derivatives (Figure 1). While the former enhances water solubility and thus facilitates the absorption and utilization of quercetin in humans by attaching sugar groups [31], the latter improves the stability, bioavailability, and overall drug properties of quercetin through methylation [32]. Therefore, in the following, we mainly discuss the therapeutic effects of quercetin and its glycosides and methyl derivatives during sepsis.

## 4. Disease Management Roles of Quercetin and Its Natural Derivatives

### 4.1. Quercetin

Multiple studies have provided direct or indirect evidence supporting the therapeutic potential of quercetin in sepsis. The mechanism of action involves the negative regulation of intracellular ROS levels and the NF-κB signaling pathway, effectively suppressing the excessive production of inflammatory factors, including TNF-α, IL-1β, and COX-2 [33,34].

In a rat model of colorectal cancer depression, quercetin exhibited antidepressant effects by reducing the expression levels of TNF-α and IL-1β in serum and the medial prefrontal lobe. It also increased the expression of BDNF (brain-derived neurotrophic factor) protein and acted on the tyrosine kinase receptor B/β-catenin axis, demonstrating its potential for treating depression [35].

Quercetin has also shown promise in reducing inflammatory factors in the hippocampus of mice in a chronic unpredictable stress model. It inhibits the secretion of COX-2, reduces the expression of nitric oxide, and restores hippocampal function, thereby exerting an antidepressant effect [36]. Studies have suggested that quercetin may inhibit the neuroinflammation–apoptosis cascade and neuronal apoptosis in the hippocampus. It has been found to inhibit the increase in TNF-α and IL-6 expression in the hippocampus of an olfactory bulb-removed rat model, indicating its potential antidepressant mechanism [37].

Research conducted by Sul et al. demonstrated that quercetin effectively reduced the levels of ROS in lung epithelial cells induced by LPS [38]. It also inhibited the nuclear translocation of NF-κB, resulting in a decrease in the levels of inflammatory cytokines, such as TNF-α, IL-1, and IL-6, which were elevated after LPS stimulation.

Furthermore, quercetin has been found to inhibit the activation of the NLR family pyrin domain containing 3 (NLRP3) inflammasome, which is involved in inflammatory responses. It reduces excessive ROS production and downregulates the expression of NLRP3, cleaved caspase-1, IL-1β, and N-gasdermin D (N-GSDMD) in macrophages. Quercetin also suppresses TLR2/myeloid differentiation factor 88 (Myd88) and p-AMP activated protein kinase (AMPK) upregulation induced by LPS/ATP in macrophages [39].

Quercetin can regulate immune responses in X-linked inhibitor of apoptosis protein (XIAP) deficiency by inhibiting IL-1β secretion and reducing IL-18 production. It has been shown to decrease levels of IL-1β and IL-18 in mice after LPS challenge [40]. Quercetin activates the Nrf2 signaling pathway, enhancing the expression and activity of Nrf2 through various mechanisms. It directly associates with Keap1, a negative regulator of Nrf2, preventing its degradation and promoting stability and functionality. Quercetin also offers protection against cytotoxicity induced by benzo[a]pyrene (B[a]P) and mitigates DNA adduct formation through aromatic hydroxylase receptor (AhR) and Nrf2 activation [41].

Severe sepsis, especially septic shock, causes extensive ischemia–reperfusion (I/R) injury [42], which often occurs in the kidney, brain, gut, lung, myocardium, retina, etc. Quercetin has been found to alleviate I/R injury by activating Nrf2 through the MAPK and phosphoinositide 3-kinase (PI3K)/ protein kinase B (PKB, also known as AKT) signaling pathways [43]. Furthermore, Li et al. demonstrated in a recent study that quercetin ameliorated neurological deficits and reduced infarct size in rats after cerebral I/R injury by regulating the PI3K/AKT/NF-κB signaling pathway and upregulating the proportion of M2 polarization of macrophages (microglia) [44]. In addition, it also reduces senescence-associated secretory phenotype (SASP) factors and senescence phenotype in nucleus pulposus cells under IL-1β treatment, thereby mitigating intervertebral disc degeneration (IDD) [45]. It upregulates the expression of Nrf2, resulting in elevated levels of antioxidant enzymes and reduced renal injury [46]. Moreover, quercetin enhances the activity of antioxidant enzymes, such as superoxide dismutase (SOD) and catalase, reducing oxidative stress and protecting against cellular damage [47].

Quercetin is considered an excellent anti-asthmatic agent. In a rat asthma model experiment, quercetin (50 mg/kg) reduced IL-6 and TNF-α and increased IL-10 in the lung tissues of asthmatic mice compared to dexamethasone (2.5 mg/kg). It also alleviates oxidative stress and inflammation, especially in tissues [48]. In conclusion, quercetin modulates the expression of proinflammatory and anti-inflammatory cytokines, promoting a more balanced immune response (Table 1). It also offers various pharmacological benefits for sepsis, improving cardiovascular health by lowering blood pressure and lipid levels. Quercetin prevents thrombosis and reduces the risk of cardiovascular disease by inhibiting clot formation, platelet aggregation, and platelet activation markers [49,50]. Furthermore, quercetin maintains the integrity of the endothelial barrier, reducing organ dysfunction, and improving outcomes in sepsis.

### 4.2. Miquelianin

Miquelianin (quercetin 3-*O*-glucuronide), a flavonoid glycoside compound, is present in plants of the Asteraceae and Moraceae families. It possesses unique characteristics, such as the addition of a glucuronic acid moiety to the quercetin molecule, which enhances its antioxidant properties. Recent studies have indicated that treatment with 0.1 μM miquelianin can effectively reduce the production of ROS and modulate various signaling pathways, including cAMP, RAS, and ERK1/2 [51]. Additionally, it has been shown to regulate the expression of genes associated with heme oxygenase 1 (HO-1), matrix metalloproteinase (MMP)-2, and MMP-9, suggesting its potential neuroprotective effects.

Studies have demonstrated that miquelianin can cross the blood–brain barrier (BBB) and reduce the production of β-amyloid (Aβ) peptides in primary cultured Tg2576 mouse models of Alzheimer’s disease [52]. Moreover, it interferes with the initial protein–protein interactions necessary for the formation of neurotoxic Aβ oligomers, specifically Aβ1-40 and Aβ1-42, further supporting its potential as a therapeutic agent. Recent research has also highlighted the role of miquelianin in promoting neurogenesis by upregulating the time-resolved Kerr rotation (TrkR) and PI3K/AKT signaling pathways. Mice supplemented with nuciferine leaf polyphenol extract (NLPE), enriched in miquelianin, exhibited increased numbers of stem cells and neurons. In vitro experiments using miquelianin-treated HT22 and SH-SY5Y cells demonstrated enhanced neurite outgrowth and elevated TrkR and PI3K/AKT levels, indicating its potential in the treatment of neurodegenerative diseases [52,53].

Furthermore, miquelianin exhibits immunomodulatory effects by inhibiting the Th2 immune response and displaying antiallergic properties. It has been shown to suppress cytokine production and IL-2 by Th2 cells while upregulating the expression of HO-1 in splenocytes [54]. In vitro experiments have demonstrated its ability to inhibit CD4+ T-cell proliferation and induce HO-1 expression through the ROS and the C-Raf–ERK1/2–Nrf2 pathway. In a mouse model of atopic dermatitis, miquelianin effectively alleviated symptoms by inhibiting the Th2 immune response.

These studies highlight the protective utility and potential medicinal value of miquelianin in various situations, including neuroprotection and immunomodulation (Table 2). Its application in the treatment of sepsis-related diseases deserves further exploration.

### 4.3. Reynoutrin

Herbs containing reynoutrin (quercetin-3-xyloside) are recognized for their potential anti-inflammatory, antioxidant, and antiviral effects, particularly against hepatitis C virus [81]. Notably, reynoutrin has displayed promising potential in improving ischemic heart failure (IHF) by targeting S100A1 [55]. In an experimental rat model of left anterior descending (LAD) ligation-induced heart failure, reynoutrin was administered at different doses, and its impact on various aspects, including cardiac function, inflammatory factors, oxidative stress, cardiomyocyte apoptosis, and myocardial fibrosis, was comprehensively evaluated (Table 2). The results revealed significant improvements in cardiac function, a reduction in the release of inflammatory factors, alleviation of oxidative stress, attenuation of cardiomyocyte apoptosis, and mitigation of myocardial fibrosis in IHF rats treated with reynoutrin [55,82].

However, it is important to note that while reynoutrin has been isolated and identified from various plants in several studies [55,83,84], there is limited evidence available to fully support its antioxidant and anti-inflammatory effects. Further investigations are required to elucidate and provide more comprehensive evidence of these effects.

### 4.4. Rutin

Rutin, also known as quercetin-3-rutinoside or sophorin, is a flavonoid compound commonly found in various plants [85]. Extensive research has been conducted on rutin, highlighting its potential as a promising active ingredient derived from medicinal plants.

One area of focus has been rutin’s beneficial effects in mitigating tau pathology (Table 2). It has been found to inhibit tau aggregation, reduce cytotoxicity caused by tau oligomers, decrease proinflammatory cytokine production, protect neuronal morphology from harmful tau oligomers, and promote the uptake of extracellular tau oligomers by microglia. In a tau-P301S mouse model, rutin demonstrated therapeutic effects by reducing pathological tau levels, regulating tau hyperphosphorylation through increased expression of PP2A, inhibiting gliosis and neuroinflammation by downregulating the NF-κB pathway, preventing microglia-mediated synapse clearance, and rescuing synaptic loss [56]. Notably, rutin is able to effectively penetrate the BBB despite its limited water solubility and low bioavailability [86]. Its impact on the brain’s nervous system is not through the regulation of the gut microbiome but rather through direct regulation of tau [56].

Rutin has also shown potential in modulating inflammation and oxidative stress responses in mice with colitis, as observed in a study by Liu et al. [57]. It significantly improved colonic permeability, as indicated by increased levels of tight junction proteins and decreased levels of FITC-dextran and endotoxin in the serum. Rutin exerted its anti-colitis effects by inhibiting the activation of the NF-κB pathway. Additionally, rutin partially restored the imbalance in the gut microbiota of mice with colitis. It increased the abundance of potential probiotics, such as *Faecalibaculum rodentium*, while reducing the levels of potentially disease-associated bacteria, such as *Romboutsia ilealis* and *Eubacterium fissicatena* group (Table 2).

Data from a double-blind, placebo-controlled trial of rutin 500 mg daily for 3 months showed significant improvements in metabolic measures, brain-derived neurotrophic factor (BDNF), and markers of inflammation and oxidative stress in patients with type 2 diabetes mellitus (T2DM) [87]. These findings highlight the diverse therapeutic potential of rutin, particularly in the context of mitigating tau pathology and modulating inflammation and oxidative stress responses. Further research is needed to fully elucidate the molecular mechanisms and clinical applications of rutin in various disease conditions.

### 4.5. Isoquercetin

Isoquercetin (quercetin 3-glucoside) is a naturally occurring polyphenol that possesses antioxidant and anti-inflammatory properties, providing protection against oxidative stress and reducing inflammation [88]. It has demonstrated potential in mitigating ethanol-induced hepatotoxicity, oxidative stress, and inflammation through the Nrf2/ARE antioxidant signaling pathway. Additionally, it regulates the expression of nitric oxide by modulating the NF-κB transcription system [58]. Isoquercetin’s high bioavailability and low cytotoxicity make it a promising candidate for preventing birth defects in diabetic pregnancies [59].

In a study by Zhou et al., isoquercetin not only enhanced spatial memory but also provided protection to hippocampal neurons in sleep-deprived mice. The study observed an increase in the levels of NLRP3 in sleep-deprived mice, which was subsequently alleviated by treatment with isoquercetin. Furthermore, isoquercetin exhibited an inhibitory effect on the upregulation of pyroptosis-related factors, such as NLRP3, caspase-1, apoptosis-associated speck-like protein containing CARD (ASC), IL-1β, IL-18, and GSDMD, induced by LPS [60].

A recent study by Zhang et al. investigated the therapeutic effects of isoquercetin on nonalcoholic fatty liver disease (NAFLD) in mice induced by a high-fat diet [61]. The study found that isoquercetin supplementation significantly regulated bile acid levels in the liver, serum, gut, and fecal samples of NAFLD mice. Additionally, it reduced the hepatic biliary sterols and triglyceride levels by 13.2% and 16.05%, respectively, in NAFLD mice. Isoquercetin achieved these effects by inhibiting FXR-Fgf15 signaling and promoting bile acid biosynthesis. It also modulated the receptors involved in bile acid transport, reabsorption, and excretion. Long-term intake of isoquercetin is suggested to have an intervention effect on the occurrence of fatty liver.

Moreover, a study on *Apocynum venetum* leaf extract (AVLE) showed that eight compounds, including isoquercetin, quercetin, rutin, and quercetin-3-*O*-glucuronide, in AVLE had protective effects against doxorubicin (DOX)-induced cardiomyocyte apoptosis [62]. AVLE administration mitigated DOX-induced oxidative stress, improved mitochondrial function, regulated apoptosis-related protein expression, and activated the AKT signaling pathway, thereby safeguarding against DOX-induced cardiotoxicity.

These findings highlight the potential therapeutic benefits of isoquercetin in various contexts, including hepatotoxicity, sleep deprivation, nonalcoholic fatty liver disease, and cardiomyocyte apoptosis (Table 2). Further research is needed to explore its mechanisms of action and clinical applications.

### 4.6. Quercetin-3-O-Sambubioside

Quercetin-3-*O*-sanburoside is a glycoside derivative of quercetin that was mentioned in a study by Wang et al., which focused on the hepatoprotective effects of *Hedyotis diffusa Willd* [63]. The researchers observed significant protective effects against liver injury and identified quercetin-3-*O*-sambubioside as the key active compound. In vitro experiments further confirmed the compound’s ability to reverse the decrease in cell viability caused by INH (isoniazid) and its effects on relevant targets.

In another study by Guo et al., the main active components of *Eucommia* male flower pollen were identified and analyzed [89]. Quercetin-3-*O*-picroside, quercetin-3-*O*-sansanoside, and quercetin-3-*O*-naringin were identified as the primary active compounds. The researchers performed purification and structure identification of these compounds and employed molecular docking methods to predict their activities. The effects of these compounds on ROS generation were evaluated using H_2_O_2_ stimulated with RAW264.7 cells as a model.

These studies highlight the potential therapeutic benefits of quercetin-3-*O*-sanburoside and related compounds, particularly in the context of hepatoprotection and antioxidant effects (Table 2). Further research is needed to explore the mechanisms of action and the potential clinical applications of these compounds.

### 4.7. Quercitrin

Quercitrin (quercetin 3-rhamnoside) is a bioflavonoid compound, shows promise in the treatment of various diseases, particularly osteoarthritis (OA) [64,90]. It has been found to reduce the expression of MMP-13 and increase collagen II expression, promoting cell proliferation and delaying the degradation of the extracellular matrix (ECM). Notably, studies conducted on chondrocytes and SW1353 cells have yielded promising results. In an animal model of OA using rats with anterior cruciate ligament transection (ACLT), quercitrin was found to activate the p-110α/AKT/mTOR signaling pathway, leading to increased bone and tissue volume and enhanced cartilage thickness in the tibial subchondral bone. These positive effects are further supported by a decrease in the OARSI score, emphasizing quercitrin’s potential for the prevention and treatment of early OA.

Quercitrin has also demonstrated hepatoprotective properties against acetaminophen (APAP)-induced liver injury [65]. This is achieved through the reduction of ROS levels, protection of the mitochondria, and the restoration of mitochondrial complex I activity. Animal models have shown that quercitrin effectively mitigated APAP-induced liver injury, resulting in improved liver function markers and reduced inflammation levels [65].

In a comprehensive study conducted by Sun et al., the molecular mechanisms of quercitrin were further explored in an inflammatory animal model induced by LPS [66]. The study focused on quercitrin’s antidepressant effects, modulation of neuroinflammation, and influence on neuroplasticity. The administration of quercitrin (10 mg/kg) resulted in rapid and sustained antidepressant effects. Within two hours, signaling molecules related to neuroplasticity in the hippocampus were upregulated, while inflammatory pathways were suppressed. Quercitrin exhibited the ability to reduce cytokine levels, restore impaired signaling, and exert anti-inflammatory effects similar to those of a PI3K inhibitor. These findings highlight the diverse therapeutic potential of quercitrin, particularly in the domains of osteoarthritis, liver injury, and neuroinflammation (Table 2).

Further investigations are warranted to deepen our understanding of the underlying mechanisms and expand the scope of the clinical applications of quercitrin.

### 4.8. Spiraeoside

Spiraeoside (quercetin 4’-*O*-glucoside) is a flavonoid glycoside compound that occurs naturally in plants. It has shown potential as a natural alternative for managing gout, a type of arthritis [91,92]. Research conducted on the marine seagrass *Halophila stipulacea* has revealed that the extract, which contains spiraeoside, reduced neutral lipid levels in zebrafish larvae [93]. In another study, spiraeoside derived from *Filipendula ulmaria* (L.) Maxim. was investigated for its inhibitory activity on monoamine oxidase (MAO), an enzyme targeted in gout treatment. Spiraeoside demonstrated inhibitory activity approximately 25 times greater than allopurinol, suggesting its potential as a natural alternative for managing gout [91].

Furthermore, spiraeoside has significant antioxidant and anti-inflammatory properties. It has shown promising inhibitory effects on aromatase, monoamine oxidase A/B, and angiotensin-converting enzymes. Additionally, spiraeoside exhibited strong inhibitory effects on the growth of HeLa cells, particularly at a concentration of 50 μg/mL [67]. Mechanistically, it was found to suppress the expression of Bcl-2 and Bid, promoting apoptosis through the activation of caspase-9/-3, and inhibited the expression of mu-2 related death-inducing gene (MUDENG).

These findings highlight the potential of spiraeoside in the management of gout and its ability to exhibit antioxidant, anti-inflammatory, and anti-cancer properties (Table 2). Further studies are necessary to explore its therapeutic applications and mechanisms in greater detail.

### 4.9. Rhamnetin

Rhamnetin is a quercetin derivative derived from *Coriandrum sativum* [94]. In the molecular structure of quercetin, the hydroxyl group’s position is substituted by a methyl group, giving rise to rhamnetin. Sepsis caused by carbapenem-resistant *Acinetobacter baumannii* (CRAB), a pathogen resistant to current antibiotics and responsible for acute lung failure, has been extensively studied. In a study carried out by Lee et al., the potential therapeutic efficacy of rhamnetin in sepsis was underscored, presenting encouraging outcomes [68]. They demonstrated that rhamnetin effectively mitigated the uncontrolled inflammatory response in sepsis by inhibiting the release of IL-6 and NO in mouse macrophages stimulated by LPS, CRAB, and *Escherichia coli* (*E. coli*) (Table 2). Additionally, in a mouse model of sepsis with CRAB or *E. coli* infection, rhamnetin administration significantly reduced the bacterial load in organs and effectively alleviated lung injury, as evidenced by levels of inflammatory factors and histological analysis of lung tissue. In the meantime, rhamnetin exhibited remarkable anti-inflammatory activity with minimal cytotoxicity.

Shatta et al. conducted a study that established a reliable in vitro model using HepG2 cells to study the effects of rhamnetin on nonalcoholic steatohepatitis (NASH) [95]. The researchers used mixtures of oleic acid (OA) and palmitic acid (PA) at different ratios and concentrations to induce NASH in the cells. They found that rhamnetin effectively modulated the molecular mechanisms of inflammation and oxidation in the HepG2 cells, leading to significant improvement in PA-induced NASH.

### 4.10. Tamarixetin

Tamarixetin (4’-*O*-methyl quercetin) is a methylated derivative of quercetin extracted from *Tamarix troupii*. It has been found to protect against cardiac hypertrophy, a compensatory response to a mechanical load that can lead to heart failure. In an anti-cardiac hypertrophy study, researchers demonstrated that tamarixetin effectively alleviated cardiac hypertrophy and ventricular dilatation in transverse aortic constriction (TAC) mice; a series of echocardiography parameters were improved, and hypertrophy markers, such as atrial natriuretic peptide (ANP), brain natriuretic peptide (BNP), and myosin heavy chain 7 (Myh7) were significantly reduced [69]. It also inhibited phenylephrine-induced hypertrophy in cardiomyocytes and reduced oxidative stress and ROS production. It suppressed the expression of apoptosis and fibrosis-related genes, reversed remodeling in the stressed heart, and prevented nuclear translocation of the nuclear factor of activated T cells (NFAT) and activation of the PI3K–AKT signaling pathway (Table 2).

In a mouse model of bacterial sepsis induced by *E. coli* K1, tamarixetin demonstrated a strong anti-inflammatory effect, leading to reduced bacterial counts and endotoxin levels. Compared to quercetin, it exhibited stronger anti-inflammatory properties in bacterial sepsis [70]. In addition, tamarixetin has shown promising effects in combating *Staphylococcus aureus* (*S. aureus*) infection, with minimal cytotoxicity. It inhibits the activity of caseinolytic protease P (ClpP), reducing the pathogen’s virulence. Tamarixetin also suppresses the transcription of genes associated with *S. aureus* pathogenicity and decreases the expression of virulence-related proteins. It inhibits hemolytic activity and enhances urease expression. In vivo studies have demonstrated its therapeutic potential in protecting against *S. aureus* pneumonia and enhancing the antimicrobial activity of cefotaxime when used in combination [71].

### 4.11. Nepetin

Nepetin (6-methoxyluteolin) is a methylated derivative of quercetin that is derived from the flowers of *Inula japonica*, Inulae flos [74]. It has shown potential in managing various diseases, including Alzheimer’s disease and T2DM [96,97]. In the context of sepsis, nepetin has demonstrated effectiveness against multiple infections. For example, Jing et al. identified nepetin (100 mg/kg) as an inhibitor of ClpP and a potential lead compound for treating methicillin-resistant *S. aureus* (MRSA) infection [72]. Nepetin effectively combated MRSA-induced pneumonia by inhibiting bacterial virulence (Table 2).

Moreover, nepetin exhibits anti-inflammatory properties by reducing the secretion and mRNA expression of pro-inflammatory cytokines, such as IL-6, IL-8, and monocyte chemoattractant protein 1 (MCP-1). This effect is achieved through the inhibition of the NF-κB and MAPK signaling pathways. Additionally, nepetin can inhibit degranulation and the production of inflammatory molecules in bone marrow-derived mast cells [73].

Nepetin also shows potential in inhibiting osteoclast differentiation, formation, and bone resorption induced by RANKL. Studies have demonstrated its protective effect against bone destruction caused by excessive osteoclast activity. This protection is attributed to the inhibition of NF-κB and MAPK signaling pathways and prevention of the TNF receptor-associated factor 6 (TRAF6)-mediated ubiquitination of Beclin-1 [75].

Furthermore, nepetin has been found to reduce inflammation and improve lung tissue in an asthma mouse model. It decreased the levels of inflammatory markers and influences immune and inflammatory responses [76].

In summary, nepetin holds promise as a natural compound with potential therapeutic applications in the management of various diseases (Table 2). Its ability to inhibit infection, modulate inflammation, and protect against bone destruction and lung injury makes it an interesting target for further research.

### 4.12. Isorhamnetin

Isorhamnetin (3’-methylquercetin) is a flavonoid compound extracted from *Hippophae rhamnoides* (L.). Lee et al. discovered that isorhamnetin can activate the cystic fibrosis transmembrane conductance regulator (CFTR), offering a potential treatment for dry eye syndrome [77]. Their investigations showed that isorhamnetin significantly enhanced CFTR chloride currents occurring in both wild-type and ΔF508-CFTR mice. Importantly, isorhamnetin had no impact on the intracellular cAMP levels or the activity of other ion channels. The topical application of isorhamnetin on mice’s ocular surface led to CFTR activation and increased tear secretion. Isorhamnetin effectively reduced ocular surface damage and the expression of inflammatory markers in an experimental dry eye mouse model. This protective effect is attributed to the activation of the AKT/sirtuin 1 (SIRT1)/Nrf2/HO-1 pathway, mitigating apoptosis, inflammation, and oxidative stress [78].

In terms of liver fibrosis, isorhamnetin has shown promise. Studies using mouse models induced by carbon tetrachloride (CCl4) or bile duct ligation (BDL) demonstrated that the oral administration of isorhamnetin effectively counteracted liver fibrosis. Its hepatoprotective properties are attributed to the inhibition of hepatic stellate cell activation, reduction of the extracellular matrix deposition, and regulation of autophagy through the modulation of TGF-β1-mediated signaling pathways [79].

Furthermore, isorhamnetin exhibits notable antiplatelet activity. It inhibits platelet aggregation triggered by collagen and TRAP-6 without causing cytotoxic effects. The antiplatelet mechanism involves the inhibition of mitochondrial function, while ROS levels remain unaffected. Isorhamnetin has demonstrated potential as an antithrombotic agent by impeding platelet deposition [80].

These findings suggest that isorhamnetin has therapeutic potential in treating dry eye syndrome, diabetes-aggravated brain injury, liver fibrosis, and platelet aggregation (Table 2). Further research is needed to explore its clinical applications and mechanisms of action in more depth.

## 5. Oral Supplementations

Quercetin and its natural derivatives have shown promising therapeutic effects. As a natural pharmaceutical standard (PS) supplement with standard management (SM), quercetin has been used in subject trials in patients with mild-to-moderate asthma attacks and rhinitis. Compared to SM alone, ingesting quercetin Phytosome^®^ (Indena Inc., Milan, Italy), a supplement that uses sunflower phospholipids, the oral absorption of quercetin increased up to 20-fold, significantly improved outcomes, increased rhinitis scores, and reduced oxidative stress [98]. This is supported by another survey. Athletes using this quercetin supplement had better improvement in symptoms, such as muscle or local pain, spasms, and reduced levels of oxidative stress compared to controls [99]. No side effects were reported [98,99], and no hepatic or renal toxicity was observed [100]. It has been claimed that the use of the supplement may be effective in reducing some of the symptoms of pollen-induced allergies [101]. It is also a potent candidate for anti-COVID-19 treatment [102,103].

Enzymatically modified isoquercitrin (EMIQ^®^, San-Ei Gen F.F.I., Inc., Toyonaka, Japan) is an isoquercetin derivative that is extracted from the flowers and buds of the Japanese pagoda tree [104]. Its bioavailability is 17-fold higher than that of quercetin aglycone, and it shows potent cardiovascular ameliorating effects in vivo. The plasma concentrations of quercetin metabolites were significantly higher after EMIQ^®^ treatment compared to placebo (*p* < 0.001) [105]. It has been found safe in many toxicity studies and has been self-affirmed as Generally Recognized as Safe (GRAS) by the FDA (GRAS Notice No. GRN000220).

Oral supplements of rutin are common. In the previously mentioned clinical trial, rutin tablets (Solgar, New Jersey, USA) were used to intervene in patients with T2DM [74]. Macuprev^®^ supplementation (Farmaplus Italia s.r.l., Rome, Italy), used in another follow-up study [106], is a vitamin complex rutin tablet; its ingredients include vitamin D3 (800 IU), vitamin B12 (18 mg), alpha-lipoic acid (140 mg), rutin (157 mg), vitamin C (160 mg), and so on.

## 6. Conclusions and Perspectives

In summary, quercetin and its natural derivatives may play a beneficial role in sepsis by reducing inflammation and oxidative stress, downregulating the expression of TLRs, regulating the immune response, and reducing organ dysfunction associated with sepsis through the PI3K/AKT/NF-κB, Nrf2/ARE, and MAPK signaling pathways (Figure 2). In addition, in practical applications, quercetin and its natural derivatives are often used as daily supplements or as adjuvant therapeutic candidates. These compounds reduce inflammation and promote healing, enhance the antibacterial effect of antibiotics, and reduce the nuisance of drugs on intestinal flora. Additionally, they are synergetic with anti-cancer drugs to enhance chemosensitivity [107,108,109,110,111]. 

However, the application of quercetin as an anti-septic agent has been largely hindered by its low water solubility, poor bioavailability, and rapid clearance, metabolism, and enzymatic degradation [27,30]. Therefore, it is important to explore strategies, such as artificial modifications [112], protective coatings, and seamless integration of quercetin and its derivatives with advanced drug delivery platforms (e.g., nanomaterials and albumin-based carriers [113,114]). For example, using amorphous chitosan oligosaccharide (COS) as a water-soluble substrate in combination with amorphous solid dispersion (ASD), quercetin-COS-ASDs showed increased oral bioavailability of 1.64–2.25-fold in rat pharmacokinetic experiments [115]. These approaches hold promise for enhancing its therapeutic efficacy, improving its systemic availability, and facilitating its successful translation into clinical practice. Quercetin and its derivatives offer numerous advantages and hold great potential for development. Future research should concentrate on preserving their natural advantages, enhancing bioavailability and absorption, and expanding their clinical applications, particularly in the field of sepsis.

## Figures and Tables

**Figure 1 biomedicines-12-00444-f001:**
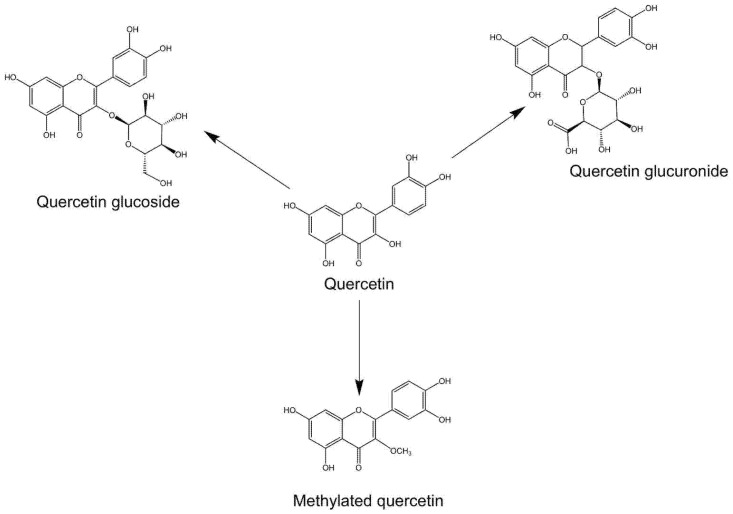
Molecular structures of quercetin, quercetin glucoside, quercetin glucuronide, and methylated quercetin.

**Figure 2 biomedicines-12-00444-f002:**
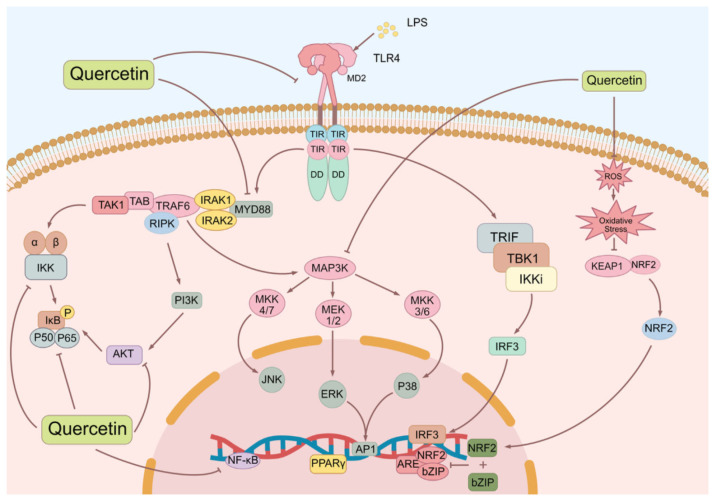
Effects of quercetin on inflammatory and oxidative pathways. AP1, activator protein 1; ARE, antioxidant response element; bZIP, basic (region-leucine) zipper; IRAK, interleukin-1 receptor-associated kinase; IκB, inhibitor of NF-κB; MD2, myeloid differentiation protein 2; RIPK, receptor-interacting protein kinase; TAB, TAK1-binding protein; TAK1, TGF-β activated kinase 1; TBK, TANK-binding kinase; TRAM, TRIF-associated adaptor molecule; TRIF, TIR-domain-containing adaptor inducing interferon-β.

**Table 1 biomedicines-12-00444-t001:** Summary of quercetin in sepsis-related in vivo/in vitro models and effective working concentrations.

Symptoms/Phenomena Associated with Sepsis	Stimulants	Main Study Subjects	Work Concentration	Ref.
ALI	LPS (10 μg/mL)	A549 cells	10 μM	[38]
Macrophage pyroptosis	LPS (10 μg/mL), ATP (5 mM)	THP-1 cells	50 μM	[39]
Immunodeficiency and hyperinflammation	In vivo: LPS (10 ng/kg) In vitro: LPS (1 μg/mL)	Human adherent monocyte; mouse BMDMs	In vivo: 50 mg/kg In vitro: 50 μM	[40]
CI/RI	the MCAO/R model	Male SD rats	50 mg/kg	[44]
Irritable bowel syndrome and IBD	Diquat (100 μM)	IPEC-1 cells	5 μM	[47]
Asthma	1% aerosolized OVA	8-week-old male Wistar rats	50 mg/kg	[48]

ALI—acute lung injury; BMDMs—bone marrow-derived macrophages; CI/RI—cerebral ischemia/reperfusion injury; SD—Sprague–Dawley; MCAO/R—middle cerebral artery occlusion/reperfusion; IPEC-1—intestinal porcine epithelial cell line 1; IBD—inflammatory bowel disease; OVA—ovalbumin.

**Table 2 biomedicines-12-00444-t002:** Summary of studies of natural quercetin derivatives with ameliorative effects.

Chemical Structure, Name, and Formula	Objectives	Subjects	Signaling Pathways/Axes	Main Outcome (Increase/Decrease)	Ref.
Miquelianin (C_21_H_18_O_1_)	Antioxidant effect	In vivo: male SD rats (275–300 g)	JNK and MAK-ERK	Decreased: ROS, cAMP and RAS, p-ERK1/2, HO-1, MMP-2, and MMP-9	[51]
	Neurodegenerative diseases	In vivo: 24-week-old male C57BL/6 mice; In vitro: HT22 and SH-SY5Y cells	TrKR and PI3K/AKT	Increased: TrKR, PI3K/AKT, neurite outgrowth levels	[52]
	Allergic diseases	In vitro: Th2 and CD4+ T cells In vivo: 6-week-old female BALB/c mice	C-Raf-ERK1/2-Nrf2	Increased: HO-1 Decreased: IL-2	[54]
Reynoutrin (C_20_H_18_O_11_)	IHF	In vivo: male SD rats (180–220 g)	NF-κB	Decreased: S100A1, MMP-2, MMP-9, p-P65, TGF-β1	[55]
Rutin (C_27_H_30_O_16_)	Nervous system inflammation and cognitive dysfunction	In vivo: 6-month-old male tau-P301S mice	NF-κB	Increased: PP2A Decreased: tau oligomers	[56]
	Colitis	In vivo: 8-week-old female C57BL/6 mice	NF-κB	Increased: TJ protein Decreased: FITC-dextran and endotoxin, *Romboutsia ilealis* and *Eubacterium fissicatena* group	[57]
Isoquercetin (C_21_H_20_O_12_)	ALD	In vitro: HepG2 cells	Nrf2/ARE, NF-κB	Decreased: NO, ROS and MDA levels, TNF-α	[58]
	Birth defects in diabetic pregnancies	In vivo: female C57BL/6 mice	NF-κB, ER-stress	Increased: SOD1 Decreased: P65, IKK, NOS2	[59]
	SD-induced neuronal injury	In vivo: C57BL/6 mice	Not mentioned	Decreased: NLRP3, caspase-1, ASC, IL-1β, IL-18, and GSDMD.	[60]
	NAFLD	In vivo: 6-week-old male C57BL/6 mice	Intestinal FXR-Fgf15 signaling	Decreased: cholesterol and triglyceride levels	[61]
	DOX-induced cardiomyocyte apoptosis	In vitro: H9c2 and HMC cells	AKT/Bcl2	Increased: LDH, ΔΨm Decreased: oxidative stress levels, AKT, Bcl-2, Bax, CytC, and pro-caspase-3	[62]
Quercetin-3-*O*-sambubioside (C_26_H_28_O_16_)	Hepatoprotective effect of HDWE	In vivo: transgenic zebrafish with a liver-specific fluorescent probe (L-FABP: EGFP)	Not mentioned	Increased: SOD, GSH Decreased: ALT, AST	[63]
Quercitrin (C_21_H_20_O_11_)	OA	In vitro: chondrocytes, SW1353 cells In vivo: male SD rats (200 ± 20 g)	P110α/AKT/mTOR	Increased: collagen II Decreased: MMP-13	[64]
	APAP-induced liver injury	In vitro: L-02 cells In vivo: BALB/c mice	Not mentioned	Increased: SOD, GSH, GSH-Px, CAT Decreased: ALT, AST, LDH, IL-6, TNF-α, ROS, MDA	[65]
	Neuroinflammation and depression	In vivo: 6–8-week-old male ICR mice (22–24 g)	PI3K/AKT/NF-κB, MEK/ERK, pCREB/BDNF/PSD95/Synapsin1	Increased: ERK signaling Decreased: IL-10, IL-1β, and TNF-α	[66]
Spiraeoside (C_21_H_20_O_12_)	Antioxidative and anti-cancer effects	In vitro: HeLa cells	Bcl-2/Bid/caspase-9/-3 pathway, cell cycle-related CDK2-cyclin E pathway	Increased: caspase-9, caspase-3 Decreased: Bcl-2 and Bid, CDK2-cyclin E, MUDENG, aromatase, monoamine oxidase A/B, and angiotensin-converting enzyme	[67]
Rhamnetin (C_16_H_12_O_7_)	Sepsis caused by CRAB and *E. coli*	In vitro: RAW 264.7 cells and HEK cells In vivo: 6-week-old female ICR mice	Not mentioned	Decreased: IL-6 and NO	[68]
Tamarixetin (C_16_H_12_O_7_)	Cardiac hypertrophy	In vitro: H9c2 cells In vivo: 8–10-week-old male C57BL/6 mice	NFAT and PI3K/AKT	Decreased: VW/BW, LW/BW, echocardiographic parameters, hypertrophic markers, ROS	[69]
	Bacterial sepsis	In vitro: BMDCs In vivo: 6-week-old female C57BL/6 mice	MAPK/JNK	Increased: IL-10 Decreased: p-JNK1, p-p38 and p-AKT, COX-2	[70]
	*S. aureus* infection	In vivo: 6-week-old female C57BL/6J mice	Not mentioned	Increased: urease Decreased: HLa and PVL proteins, ClpP activity, *hla*, *agr*, *RNAIII*, *pvl*, *PSM-α* and *spa* genes	[71]
Nepetin (C_16_H_12_O_7_)	MRSA infection	In vivo: 7-week-old female C57BL/6J mice	Not mentioned	Decreased: thermal stability of ClpP	[72]
	RPE	In vitro: ARPE-19 cells	NF-κB and MAPKs	Decreased: IL-6, IL-8, and MCP-1	[73]
	Inflammation and allergy	In vitro: BMMCs In vivo: male BALB/c mice	AKT/NF-κB/COX-2, and IGE/Ag	Decreased: COX-2, PLCγ1, cPLA2, LTC4, PGD2, Ca^2+^	[74]
	Inflammatory osteolysis	In vitro: osteoclasts In vivo: 6-8-week-old male BALB/c mice	NF-κB and MAPK	Decreased: TRAF6- dependent ubiquitination of Beclin-1	[75]
	CFD-induced pneumonia and asthma	In vitro: MH-S cells In vivo: 6-week-old C57BL/6 mice	NF-κB and MAPK, the localization of IRAK-1	Decreased: NO, INOS, COX-2, IL-1β, IL-6, and TNF-α, ADMA, SDMA	[76]
Isorhamnetin (C_16_H_12_O_7_)	DED	In vitro: FRT cells, T84 cells, HEK-293T cells, CHO cells, human CorE and CorjE cells In vivo: 6-week-old ICR mice (30 g)	PI3K/AKT and NF-κB	Increased: CFTR activity and tear secretion Decreased: IL-1β, IL-8, and TNF-α	[77]
	Diabetes-associated cerebral I/R injury	In vitro: HT22 cells	AKT/SIRT1/Nrf2/HO-1	Decreased: SIRT1, Nrf2, HO-1, p-AKT	[78]
	Liver fibrosis	In vivo: 8-week-old male C57 mice (22–24 g)	TGF-β1-mediated SMAD3 and p38-MAPK	Increased: PPAR-γ, MMP-2 Decreased: TGF-β1, SMAD3, ALT, AST, Collagen I and III, *α*-SMA, Beclin-1, LC3, TIMP-1	[79]
	Antiplatelet and antithrombotic effects	In vitro: human peripheral blood	Not mentioned	Increased: Ca^2+^ Decreased: platelet ATP levels, ΔΨm	[80]

ALD—alcohol-related liver disease; BMMC—bone marrow mononuclear cell; CHO—Chinese hamster ovary; ConjE—conjunctival epithelial; CorE—corneal epithelial; DED—dry eye disease; FRT—fisher rat thyroid; ICR—Institute of Cancer Research; LAD—left anterior descending; LW/BW—lung weight/body weight ratio; MMP—matrix metallopeptidase; MUDENG—mu-2 related death-inducing gene; NFAT—nuclear factor of activated T cells; RPE—retinal pegment epitheliitis; S100A1—S100 calcium-binding protein A1; SD—sleep deprivation; SPF—specific-pathogen-free; TJ—tight junction; VW/BW—ventricular weight/body weight ratio; ΔΨm—mitochondrial membrane potential.

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
