# Peer review of "Unlocking the Potential: Quercetin and Its Natural Derivatives as Promising Therapeutics for Sepsis"

_biomedicines, 2024, doi:10.3390/biomedicines12020444_

Round 1

Reviewer 1 Report

Comments and Suggestions for Authors

The following comments should be considered before publication:

Figure 1 contains several errors, as follows:

Quercetin glycoside...change to Quercetin glucoside.

Mythylated quercetin...change to...Methylated quercetin.

Please revise and correct the structure of the glucose moiety in quercetin glucoside as an oxygen of the pyran ring is missing. Also, use a solid wedge bond to indicate beta-linked glucose.

Please revise and correct the structure of the methoxy substitution at C-3 of methyl quercetin.

Comments on Table 1:

I suggest that the first column, “Type of derivatives” be deleted and replaced with a side title (i.e., as a row). This will provide a space for expanding the other columns and allow increasing the size of the structures’ images.

All sentences within Table 1 should be capitalized.

Generally, in vitro and in vivo should be in italic font (in whole MS).

What is the significance of the bold font in Table 1? Please justify in a footnote or remove the bold font style.

Line 112, Lines 115-116: Please delete the steam distillation method since it is not applied for quercetin or its derivatives because they are non-volatile compounds.

Line 194, 207: remove extra space at the ends of the statements.

Line 213: In a similar way to other mentioned compounds, I suggest adding quercetin-3-xyloside to define this structure.

Line 278: The letter “L.” for the abbreviated scientist name, should be written in regular font, not italic.

Lines 278-279: Moreover, isoquercetin and its derivatives from Apocynum venetum L. (AVLE) demonstrated protective effects against doxorubicin.... etc. This statement should be modified to indicate that “Apocynum venetum L. (AVLE), a plant rich in isoquercetin and its derivatives, demonstrated protective effects against doxorubicin.... As the study mentioned focused on the effects of the extract (i.e., AVLE) rather than isolated compounds.

Section 4.6., Lines 287-301: Generally, the oxygen atom “O” linkage in the mentioned glycosides should be in italic font. This should be revised in the whole MS.

Section 4.7., Lines 306-315: Please revise the font style.

Lines 341-342: Filipendula ulmaria.... italic font should be used for all binomial names (i.e., Genus species). Similarly in lines 358, 365, and 390 (Please revise). Also, binomial names should be mentioned once as full genus and species names, then abbreviated later as Escherichia coli.... E. coli and Staphylococcus aureus.... Staph. aureus etc.

For other comments, see the attached pdf.

Comments on the Quality of English Language

No detected issues

Author Response

1.Summary

2.Point-by-point response to Comments and Suggestions for Authors

  1. Figure 1 contains several errors, as follows: Quercetin glycoside...change to Quercetin glucoside. Mythylated quercetin...change to...Methylated quercetin. Please revise and correct the structure of the glucose moiety in quercetin glucoside as an oxygen of the pyran ring is missing. Also, use a solid wedge bond to indicate beta-linked glucose. Please revise and correct the structure of the methoxy substitution at C-3 of methyl quercetin.

Reply: Thank you for pointing that out. The title and compound structures in Figure 1 have been corrected accordingly.

Figure 1. Molecular structures of quercetin, quercetin glucoside, quercetin glucuronide, and methylated quercetin.

  1. Comments on Table 1: I suggest that the first column, “Type of derivatives” be deleted and replaced with a side title (i.e., as a row). This will provide a space for expanding the other columns and allow increasing the size of the structures All sentences within Table 1 should be capitalized. Generally, in vitro and in vivo should be in italic font (in whole MS). What is the significance of the bold font in Table 1? Please justify in a footnote or remove the bold font style.

Reply: I appreciate your feedback on improving Table 1's structure. To increase readability, we removed the 'derivative classification' in the first column and added the molecular formula for each compound. We have also corrected writing errors in Table 1.

  1. Line 112, Lines 115-116: Please delete the steam distillation method since it is not applied for quercetin or its derivatives because they are non-volatile compounds.

Reply: Thank you for your kind remind. The citation error has been rectified in Lines 128-132.

  1. Line 194, 207: remove extra space at the ends of the statements.

Reply: Thank you for your revision. We have removed excess spaces and rechecked the manuscript.

  1. Line 213: In a similar way to other mentioned compounds, I suggest adding quercetin-3-xyloside to define this structure.

Reply: Thank you for your useful suggestion. To make the description easier to understand, we added compound aliases for each derivative.

  1. Line 278: The letter “L.” for the abbreviated scientist name, should be written in regular font, not italic.

Reply: I appreciate your professional modification. We have corrected such kind of typographical errors in the full text.

  1. Lines 278-279: Moreover, isoquercetin and its derivatives from Apocynum venetum L. (AVLE) demonstrated protective effects against doxorubicin.... etc. This statement should be modified to indicate that Apocynum venetum L. (AVLE), a plant rich in isoquercetin and its derivatives, demonstrated protective effects against doxorubicin.... As the study mentioned focused on the effects of the extract (i.e., AVLE) rather than isolated compounds.

Reply: Thank you for your advice. We have revised this part at Lines 311-316 accordingly.

  1. Section 4.6., Lines 287-301: Generally, the oxygen atomOlinkage in the mentioned glycosides should be in italic font. This should be revised in the whole MS.

Reply: Thank you for your comments. We checked and corrected such formatting errors in the manuscript and marked them in red.

  1. Section 4.7., Lines 306-315: Please revise the font style.

Reply: We apologize for the formatting error. The necessary revisions have been made in the first paragraph in Section 4.7.

Lines 341-342: Filipendula ulmaria.... italic font should be used for all binomial names (i.e., Genus species). Similarly in lines 358, 365, and 390 (Please revise). Also, binomial names should be mentioned once as full genus and species names, then abbreviated later as Escherichia coli.... E. coli and Staphylococcus aureus.... Staph. aureus etc.

Reply: Thank you for your feedback. We have made the necessary revisions as per your suggestions.

Reviewer 2 Report

Comments and Suggestions for Authors

The authors may change the title since the manuscript not only focused on quercetin but also discussed other bioactive compounds. try to alter the title

The abstract needs substantial revision, the present form is less informative.

The majority of the data in this review can be found easily in previously published papers or books. On the other hand, the scientific quality of this review is very low and author need to increase the novelty or significance of this review

The highlights of this review should be revised. The main findings should be mentioned.
The English language should be improved.

The main gaps in the previous studies should be discussed.

The novelty of the present review is not clear.

There are lots of general sentences in sections 3, and 4.

The author may incorporate more informative tables about possible mechanisms of each bioactive molecules in various biomedical applications/potentials

Need valid references "As one of the widely occurring polyphenols, quercetin and the derivatives can be extracted readily with methods like solvent extraction, steam distillation, and chromatography."

in section 4, selective bioactive molecules and natural source-related information are lacking

Briefly state the commercially available Quercetin and their real-world applications

The final outcome of this review, mentioned in the conclusion section, is not very clear.

The author may also improve the future perspective or suggestions of this review

Comments on the Quality of English Language

manuscript needs substantial revision

Author Response

1.Summary

2.Point-by-point response to Comments and Suggestions for Authors

  1. The authors may change the title since the manuscript not only focused on quercetin but also discussed other bioactive compounds. try to alter the title.

Reply: Thank you for your constructive suggestions. We have modified the title to 'Unlocking the Potential: Quercetin and its Natural Derivatives as Promising Therapeutics for Sepsis' accordingly.

  1. The abstract needs substantial revision, the present form is less informative.

Reply: The Abstract has been rewritten. Hopefully now it provides more information to the readers.

  1. The highlights of this review should be revised. The main findings should be mentioned.

Reply: Thank you for your proposal. We have redivided the structure of the article and rewritten the section 6 (Conclusions and perspective). We summarize the main conclusions of the manuscript at Lines 512-520.

  1. The English language should be improved.

Reply: Thanks for the suggestion. The manuscript has been polished by an English language editing companyrecommended by the MDPI. We have also carefully checked the whole manuscript to correct grammar/typo errors.

  1. The main gaps in the previous studies should be discussed.

Reply: In response to your suggestion, we have added Section 5 ‘Oral supplementations’ to provide current information about the oral-supplemental products. In addition, the major deficiencies of these compounds have been described in Section 6 (Lines 521-523), which require further exploration and enhancement through systematic food and drug delivery systems.

  1. The novelty of the present review is not clear.

Reply: Thank you for your advice. We have added the core objectives and intentions of this review to the manuscript (Line 48-53), hoping to convey the original intention of our writing.

  1. here are lots of general sentences in sections 3, and 4.

Reply: Thank you for your modification suggestions. We have linguistically refined the full text of the manuscript.

  1. The author may incorporate more informative tables about possible mechanisms of each bioactive molecules in various biomedical applications/potentials.

Reply: We appreciate your feedback. We have revised Table 1 to provide a concise summary of the targets and major signaling pathways of quercetin derivatives as mentioned in recent publications. Additionally, we have added Table 2 to encompass the specific applications and efficacy measurements of quercetin across various experimental models.

  1. Need valid references "As one of the widely occurring polyphenols, quercetin and the derivatives can be extracted readily with methods like solvent extraction, steam distillation, and chromatography."

Reply: Thank you for your modification suggestions. We rechecked the manuscript and references, excised incorrect descriptions, and added appropriate citations.

  1. in section 4, selective bioactive molecules and natural source-related information are lacking.

Reply: Thank you for your suggestions. We have added the information of natural sources of some derivatives in section 4.

  1. Briefly state the commercially available Quercetin and their real-world applications.

Reply: We appreciate your valuable advice. In Section 5 (Oral supplementations), we have included details about commercially available oral supplements of quercetin and its derivatives, relevant clinical trials, and reports. Additionally, we have addressed the extent to which these supplements have been enhanced in terms of bioavailability.

  1. The final outcome of this review, mentioned in the conclusion section, is not very clear.

Reply: We appreciate your valuable input. As per your advice, we have extended the initial ending section to Section 5 to encompass the description of available market information, and Section 6, specifically Lines 512-520, to provide a summary of available conclusions. Furthermore, Lines 523-529 now address the primary limitations of this class of compounds and potential avenues for future research and application. Thank you for guiding us in enhancing the content.

  1. The author may also improve the future perspective or suggestions of this review.

Reply: Thank you for your advice. We reconsidered and wrote Section 6 (Conclusions and perspective), in which Line 523-533 was used to look into the future optimization direction for the development of such compounds.

Reviewer 3 Report

Comments and Suggestions for Authors

The subject is interesting, just for me seems to be a difference between the title and the content. In title is mentioned the sepsis, but are very few data about the use of quercetin and its derivatives in sepsis. It is not highlighted enough how are linked the compounds to the use in sepsis or a basic prove of the positive effect of these compounds in case of sepsis.

In other hand it is a good review of recent literature, but it does not mentioned the method in that the data were collected. I wished to read also about a comparison between the compounds bioavailability. 

There are the same information in table and text. 

The text contains 101 references, but in list are 102.

All Latin words must be written with italic letters.

I propose to authors to reconsider all work and re-organize it to increase its scientific value.

Comments on the Quality of English Language

The English is fine, some minor typing errors must be corrected.

Author Response

1.Summary

2.Point-by-point response to Comments and Suggestions for Authors

  1. The subject is interesting, just for me seems to be a difference between the title and the content. In title is mentioned the sepsis, but are very few data about the use of quercetin and its derivatives in sepsis. It is not highlighted enough how are linked the compounds to the use in sepsis or a basic prove of the positive effect of these compounds in case of sepsis.

Reply: Thank you for your feedback. We acknowledge the limited direct research on the use of quercetin and its derivatives in sepsis. However, there is promising research on its potential application in other inflammation-related diseases, which is the basis for our review. We hope to raise awareness and further explore its potential use in sepsis treatment. Accordingly, we have modified the title to 'Unlocking the Potential: Quercetin and its Natural Derivatives as Promising Therapeutics for Sepsis'. We also added a table (Table 2) to address the points you raised.

  1. In other hand it is a good review of recent literature, but it does not mentioned the method in that the data were collected.

Reply: Thanks for the useful suggeations. We have updated the abstract to include a description of the literature search method.

  1. I wished to read also about a comparison between the compounds bioavailability.

Reply: In response to your valuable input, we have incorporated reports of clinical trials and investigations pertaining to presently accessible oral quercetin and its derivatives in the manuscript. These additions can be found in Section 5, specifically within Lines 491-501. Furthermore, we have included an analysis of the side effects and bioavailability of these compounds.

  1. There are the same information in table and text.

Reply: We have made revisions to Table 1 in order to improve the accessibility of information for our readers. The information in the table has been refined, with a reduction in duplication and redundant words in the main text. Additionally, based on valuable feedback, we have updated the object column to include the type of experiment object that was not previously mentioned in the main text. These changes aim to enhance the clarity and usefulness of the table for our audience. Thank you for your helpful advice.

  1. The text contains 101 references, but in list are 102.

Reply: Thank you for your kind remind. We have corrected the formatting errors and annotated them in the manuscript.

  1. All Latin words must be written with italic letters.

Reply: We have thoroughly reviewed the entire text and made the necessary adjustments.

Round 2

Reviewer 2 Report

Comments and Suggestions for Authors

The authors addressed each comment, and manuscript quality is now improved and suitable for publication